# Dietary Replacement of Soybean Meal with *Zanthoxylum bungeanum* Seed Meal on Growth Performance, Blood Parameters, and Nutrient Utilization in Broiler Chickens

**DOI:** 10.3390/ani14101420

**Published:** 2024-05-09

**Authors:** Xing Chen, Yang Li, Aijuan Zheng, Zedong Wang, Xu Wei, Shuzhen Li, Adanan Purba, Zhimin Chen, Guohua Liu

**Affiliations:** 1Key Laboratory for Feed Biotechnology of the Ministry of Agriculture and Rural Affairs, Institute of Feed Research, Chinese Academy of Agriculture Sciences, Beijing 100081, China; cx18222052919@163.com (X.C.); zhengaijuan@caas.cn (A.Z.); wangzedong@caas.cn (Z.W.); 17332337527@163.com (X.W.); lishuzhen1543@163.com (S.L.); mhdadananpurba@usu.ac.id (A.P.); chenzhimin@caas.cn (Z.C.); 2Beijing Dabeinong Technology Group Co., Ltd., Beijing 100194, China; liyang8906@163.com

**Keywords:** *Zanthoxylum bungeanum* seed meal, growth performance, blood indicator, nutrient utilization, broiler chicken

## Abstract

**Simple Summary:**

The main objective of this research is to find an effective way to reduce the reliance of broiler chickens on soybean meal. ZBM is a byproduct obtained by squeezing *Zanthoxylum bungeanum*, which contains high levels of protein, fat, and minerals. Currently, there is no existing research on the application of ZBM in broiler chickens. Therefore, this experiment aims to validate the effectiveness of using ZBM as a substitute for soybean meal in broiler chicken feed. The validation process involves two parts: in vitro biomimetic digestion experiments and broiler production experiments. The results demonstrated that the utilization of 5–10% ZBM led to notable enhancements in the growth and development of broilers. Additionally, it was observed that digestibility and utilization of nutrients were increased. Based on our research findings, we strongly believe that ZBM has the potential to be a suitable replacement for soybean meal in broiler feed.

**Abstract:**

*Zanthoxylum bungeanum* seed meal (ZBM), a novel plant protein raw material, has shown promising potential in enhancing the growth of broiler chickens as a substitute for soybean meal (SBM) in feed. In the artificial digestive experiment of vitro experiments, the digestibility of ZBM and SBM were assessed using the SDS-III Single Stomach Animal Biometric Digestion System. Subsequently, 180 1-day old AA chicks were divided into three groups for in vivo experiments: corn–soybean-meal-based diet (CON group); ZBM replacing 5% soybean meal in the basal diet (ZBM-1 group); ZBM replacing 10% soybean meal in the basal diet (ZBM-2 group). The experiment period lasted for 42 days. Compared to SBM, ZBM demonstrated higher crude protein content, dry matter digestibility, and extracorporeal digestible protein. Compared with the CON group, the broilers in the ZBM-2 group showed improved ADG and ADFI during the 1–21 d, 22–42 d, and 1–42 d periods (*p* < 0.05). Furthermore, the ZBM groups exhibited significant increases in slaughter performance compared with the CON group (*p* < 0.05). The substitution of ZBM for SBM also leads to a significant reduction in serum enzyme indicators (*p* < 0.05). Additionally, the lipoprotein and total cholesterol of the ZBM groups were significantly lower than those of the CON group (*p* < 0.05). Substituting SBM with ZBM significantly enhances the activity of superoxide dismutase and the content of immunoglobulin G in broiler serum, while reducing the content of malondildehyde (*p* < 0.05). The ZBM groups showed significantly higher utilization of dry matter, crude protein, and energy compared with the CON group (*p* < 0.05). In conclusion, the study confirmed that the substitution of SBM with 5–10% ZBM in broiler diets has a significant positive effect on growth, development, antioxidant capacity, immune function, and nutrient utilization. This study not only provides a theoretical foundation for the utilization of ZBM in broiler diets but also offers an effective approach for reducing reliance on soybean meal.

## 1. Introduction

Soybean meal is widely used as a protein source in poultry nutrition due to its easily obtainable nature and relatively balanced amino acid content [1,2]. In recent years, the global food crisis has been exacerbated by multiple factors, including economic shocks, conflicts and insecurities, climate change, and extreme weather events [3,4]. Affected by these challenges, soybean meal prices are unstable and the supply–demand relationship is imbalanced [5,6]. The scarcity of soybean meal and the significant increase in feed prices have emerged as crucial factors that hinder the sustainable development of the industry [7,8]. Therefore, the development and application of new protein resources to replace soybean meal has become an urgent task.

*Zanthoxylum bungeanum* (ZB) is a deciduous perennial plant that belongs to the rutaceae family and the zanthoxylum genus [9,10]. It is distributed extensively in tropical and subtropical regions, such as India, Bangladesh, Indonesia, China, and Malaysia [11]. Over 140 chemical components, such as alkaloids, terpenoids, flavonoids, polyunsaturated fatty acids, and a small number of inorganic elements, have been isolated and identified from *Zanthoxylum bungeanum* [12,13]. These components exhibit various effects including analgesic, anti-inflammatory, antifungal, antioxidant and antitumor effects [14]. *Zanthoxylum bungeanum* seeds refer to the seeds of mature *Zanthoxylum bungeanum* that are extracted from the peel after harvesting [15]. These seeds are known for their high oil content, primarily consisting of unsaturated fatty acids, which makes them a desirable choice as an edible vegetable oil [16].

*Zanthoxylum bungeanum* seed meal (ZBM) is a byproduct obtained by pressing *Zanthoxylum bungeanum* seeds, which are rich in nutrients such as protein, fat, and minerals [17]. However, the potential of utilizing ZBM as a viable alternative for soybean meal in broiler feed has not yet been explored. Therefore, this experiment verifies the effects of applying ZBM as a substitute for soybean meal in feed through two experimental parts: an in vitro biomimetic digestion experiment and a broiler production experiment. The results provide a theoretical foundation for the utilization of ZBM and offer a new approach to reducing and replacing soybean meal.

## 2. Materials and Methods

### 2.1. Ethics Statement

The research was licensed by the ethical approval of the Animal Care and Use Committee of the Institute of Feed Research of the Chinese Academy of Agricultural Science (Statement No. AEC-CAAS-20191106, Beijing, China).

### 2.2. Animals and Diets

A total of 180 one-day-old Arbor Acres (AA) broiler chicks were randomly divided into three treatments (6 replicates/group and 10 chicks/replicate). The initial body weight was similar across the replicates; the chicks were housed in three-tier metal cages (140 cm width × 70 cm depth × 40 cm height). The control group (CON group) of chicks was provided with a basal diet formulated to meet or exceed the standards set by NRC (1994) [18]. The ZBM-1 group and the ZBM-2 group were fed a diet with *Zanthoxylum bungeanum* seed meal replacing 5% and 10% of the soybean meal in the basal diet, respectively. The *Zanthoxylum bungeanum* seed meal (ZBM) was kindly provided by Shaanxi Weikang Biotechnology Co., Ltd. (Hancheng, China). The amino acid contents of SBM and ZBM are shown in Table 1. Detailed summaries of the compositions of the basal and the 2 experimental diets tested are provided in Table 2. The metabolizable energy of the feed was derived based on the feed components listed in the nutritional values. The crude protein content in the feed was analyzed using a Dumas nitrogen analyzer (D50, Haineng Future Technology Group Co., Ltd., Shanghai, China). Calcium and phosphorus content were determined according to the Chinese national standard methods (GB/T 6436-2018 and GB/T 6437-2018, respectively). The content of lysine, methionine, threonine, and cysteine in each group of feed was measured using the Chinese national standard method (GB/T 18246-2019), and tryptophan content was determined using the Chinese national standard method (GB/T 15400-2018). The above amino acid contents were analyzed in a fully automated amino acid analyzer (Hitachi L-8900, Tokyo, Japan). The premix, feeding management, and vaccination of broiler chickens were the same as previously published [19].

### 2.3. In Vitro Digestibility

In the in vitro experiments, based on “Biomimetic Digestion Measurement Procedure for Total Digestible Carbohydrates in Chicken Feed”, the DM and CP digestibility of SBM and ZBM were determined by the SDS-III Single Stomach Animal Biomimetic Digestion System (Hunan Zhongben Intelligent Technology Development Co., Ltd., Changsha, China). The SDS-specific digestive enzyme reagent (specific to chicken digestion) is composed of pepsin (Sigma P7000, St. Louis, MO, USA, 387.5 KU–250 mL), trypsin (Amresco 0785, Solon, OH, USA, 13.55 KU–25 mL), chymotrypsin (Amresco 0164, 3.11 KU–25 mL), and amylase (Sigma A3306, 110.40 KU–25 mL). The preparation method for simulating gastric juice (1550 U/mL) was as follows: dissolve pepsin in hydrochloric acid with pH = 2.0 and dilute to a final volume of 250 mL. The preparation method for simulating small intestine fluid (including 401.46 U/mL amylase activity, 49.28 U/mL trypsin activity, 11.31 U/mL chymotrypsin activity) was as follows: dissolve amylase, trypsin, and chymotrypsin in a deionized water and adjust the volume to 25 mL.

### 2.4. Growth Performance

The experiment consisted of two nutrition stages: the starter stage (days 1–21) and the grower stage (days 22–42). Throughout these stages, the average daily gain (ADG) and average daily feed intake (ADFI) of the broilers were measured. These measurements were also taken for the entire duration of the experiment (1–42 days). The feed-to-gain ratio (F/G) was calculated, and the number of broiler deaths was recorded and they were weighed daily to correct the F/G.

### 2.5. Slaughter Performance

On the 42nd day of the experimental period, 12 broiler chickens with similar weights (2 chickens per replicate) were chosen from each treatment. These selected chickens underwent a 12-h fasting period and were allowed to drink freely. The purpose of this was to assess carcass characteristics, including dressed percentage (DP), half-eviscerated weight rate (HEWR), eviscerated weight rate (EWR), pectoral muscle rate (PMR), leg muscle rate (LMR), and abdominal fat rate (AFR).

### 2.6. Blood Sample Collection and Analysis

At 21 and 42 days of age, two broilers with similar weights were selected in each replicate for blood collection from the brachial wing vein after a 12-h fasting period. The determination indicators and method for serum were the same as those previously published [19].

### 2.7. Apparent Utilization Rate of Feed Nutrients

The indicator fecal collection method was employed to determine the apparent utilization rate of DM, CP, and energy (EN) in two stages: at the ages of 18–21 days (acid-insoluble ash method, AIA method) and 39–42 days (titanium dioxide method, 0.4% TiO_2_ method). The content of CP was determined according to a Dumas nitrogen analyzer (D50, Haineng Future Technology Group Co., Ltd., Shanghai, China) and the EN was determined utilizing oxygen bomb calorimeter (IKA-C3000, Staufen, Germany). Prior to the experiment, the feed trough was thoroughly cleaned and the animals were fasted for 12 h. Throughout the metabolic experiment, fecal samples were collected daily, and the daily feed intake and leftover amounts were recorded.

### 2.8. Statistical Analysis

Shapiro–Wilk and Levene tests were used to verify the normal distribution and homogeneity of the variances of the data. The experiment data were analyzed using one-way ANOVA and the general linear model (GLM) in SAS 9.4 (SAS Institute Inc., Cary, NC, USA) to evaluate the replacement of soybean meal in the diet with ZBM. Variations among the treatments (CON, ZBM-1, and ZBM-2 groups) were compared using Tukey’s multiple range tests. The experimental data were organized into figure format using GraphPad Prism 8 software (GraphPad Software, San Diego, CA, USA). The results were presented as the mean and standard error of the mean (SEM). The differences between the treatments were displayed by * *p* < 0.05.

## 3. Results

### 3.1. Artificial Digestive Experiment In Vitro

Figure 1 presents the in vitro digestibility results for SBM and ZBM. The indicators of CP, DM-D, and ED-P were higher than ZBM compared with SBM. On the other hand, the moisture content and CP-D of ZBM showed a decrease compared with SBM.

### 3.2. Growth Performance

As shown in Figure 2, in the initial phase (1–21 d), the ADG and ADFI of the ZBM-1 and ZBM-2 groups were significantly higher than those of the CON group (*p* < 0.05). There were no significant changes in F/G between the CON, ZBM-1, and ZBM-2 groups (*p* > 0.05). In the grower phase (22–42 d), the ADG of the ZBM-2 group was significantly higher than that of the CON group (*p* < 0.05). Additionally, there was no significant alteration in ADFI between the CON group and the ZBM-1 group (*p* > 0.05), but a significantly lower ADFI level was observed when compared with the ZBM-2 group (*p* < 0.05). The F/G of the ZBM-2 group was higher than that of the ZBM-1 group (*p* < 0.05); there was no significant difference between the CON group (*p* > 0.05). Throughout the entirety of the experiment, the ADG and ADFI of the ZBM-2 group were significantly higher than those of the CON and ZBM-1 groups (*p* < 0.05), and there were no significant alterations between the CON and ZBM-1 groups (*p* > 0.05). Additionally, there were no significant differences in the F/G between the CON, ZBM-1, and ZBM-2 groups (*p* > 0.05).

### 3.3. Slaughter Performance

As shown in Figure 3, the DP and PMR of the CON group were lower than those of the ZBM-1 and ZBM-2 groups (*p* < 0.05), but there was no significant difference between the ZBM-1 and ZBM-2 groups (*p* > 0.05). Additionally, the HEWR and EWR of the ZBM-2 group were higher compared with the CON and ZBM-1 groups (*p* < 0.05); meanwhile, those of the CON group were lower than those of the ZBM-1 group (*p* < 0.05). Similarly, the LMR of the ZBM-2 group was higher than that of the CON group (*p* < 0.05), but there was no significant change to the ZBM-1 group (*p* > 0.05). On the contrary, compared with the CON group, the ZBM-1 and ZBM-2 groups in the AFR was decreased significantly (*p* < 0.05). However, there were no significant differences between the ZBM-1 and ZBM-2 groups (*p* > 0.05).

### 3.4. Serum Biochemical Parameters

As shown in Figure 4, in the 21-day period, the ALT and AST indexes of the ZBM-2 group were significantly lower than those of the CON group (*p* < 0.05); meanwhile, there were no significant alterations observed in the ZBM-1 group (*p* > 0.05). On the other hand, the TCHO contents of both the ZBM-1 and ZBM-2 groups were significantly lower than that of the CON group (*p* < 0.05), with no significant difference between the ZBM-1 and ZBM-2 groups (*p* > 0.05). However, no significant differences were observed in the serum TP, ALB, GLB, BUN, ALP, and TG levels of the 21-day broiler chickens (*p* > 0.05). At 42 d, the ALB and GLB contents of the ZBM-1 group were significantly higher than those of the CON group (*p* < 0.05). However, there were no significant differences in ALB and GLB between the ZBM-1 and ZBM-2 groups (*p* > 0.05). Furthermore, there was no significant effect on ALT between the CON and ZBM-2 groups (*p* > 0.05), but both were higher than those in the ZBM-1 group (*p* < 0.05). Similarly, the AST contents of the ZBM-1 group were higher than that of the CON group (*p* < 0.05). Regarding ALP, the ZBM-2 group had higher contents than both the CON and ZBM-1 groups (*p* < 0.05), and the ZBM-1 group had higher contents than the CON group (*p* < 0.05). In terms of lipid indicators at 42 d, the LP and TCHO contents of the CON group were higher than those of the ZBM-1 and ZBM-2 groups (*p* < 0.05); meanwhile, there were no significant differences between the ZBM-1 and ZBM-2 groups (*p* > 0.05).

### 3.5. Serum Antioxidant Capacity

As shown in Figure 5, in 21 d, the SOD activity of ZBM-1 group was higher than that of the CON group (*p* < 0.05). There were no significant effects in T-AOC, MDA content, or GSH-Px activity between the CON, ZBM-1, and ZBM-2 groups (*p* > 0.05). In 42 d, the SOD activity of the ZBM-1 and ZBM-2 groups was higher than that of the CON group (*p* < 0.05); meanwhile, there were no significant differences between the ZBM-1 and ZBM-2 groups (*p* > 0.05). On the other hand, the MDA content of the ZBM-1 and ZBM-2 groups was lower than that of the CON group (*p* < 0.05), and there were no significant observed effects between the ZBM-1 and ZBM-2 groups (*p* > 0.05).

### 3.6. Serum Immunoglobulin Content

As shown in Figure 6, in 21 d, there were no significant alterations in IgA, IgM, or IgG content between the CON, ZBM-1, and ZBM-2 groups (*p* > 0.05). In 42 d, compared to the CON group, the IgG content of the ZBM-1 and ZBM-2 groups was increased (*p* < 0.05), and there were no significant changes between the ZBM-1 and ZBM-2 groups (*p* > 0.05).

### 3.7. Apparent Utilization Rate of Feed Nutrients

As shown in Figure 7, in 21 d, the DM, CP, and EN utilization of the ZBM-2 group were significantly higher than those of the CON and ZBM-1 groups (*p* < 0.05). In 42 d, there were no significant alterations in DM or EN utilization between the CON and ZBM-2 groups (*p* > 0.05), but both were lower than those in the ZBM-1 group (*p* < 0.05). Additionally, the CP utilization of the ZBM-2 group was lower than that of the ZBM-1 group (*p* < 0.05), but no significant differences were found in comparison with the CON group (*p* > 0.05).

## 4. Discussion

The rise in prices of soybean meal and the challenges in sourcing it have significantly impeded the progress of animal husbandry [8,20]. Thus, the current trend and focus of research is to develop new protein feed raw materials that can replace soybean meal. Our observations suggest that ZBM may be a viable alternative to soybean meal, as evidenced by the positive impact it has on the growth performance, antioxidant capacity, immune function, and nutrient utilization of broiler chickens.

Prior to this, several articles have examined the impact of substituting soybean meal with various plant protein sources on the growth and development of broilers. For example, replacing 50% of soybean meal in the diet of broiler chickens with linseed meal does not have negative impact on feed intake, feed conversion rate, or mortality rate [21]. The production performance of broilers was significantly improved when perilla seed meal replaces 1–2% soybean meal in the diet [22]. In addition, substituting soybean meal in the diet with plant protein raw materials like fermented rapeseed meal, raw chickpea, sweet almond meal, raw and fermented rapeseed cake, and sprouted whole pearl millet can also contribute to maintaining or even enhancing the growth and development of broilers [23,24,25,26,27]. In the experiment, the substitution of 5–10% soybean meal with ZBM significantly improved the ADG and ADFI of broilers, which supports the feasibility of using ZBM in broiler diets. On the other hand, slaughtering performance is a crucial indicator for evaluating the quality of meat from livestock and poultry, providing insights into the distribution of different tissue parts in the overall mass and reveals variations in nutrient deposition among these parts [28,29]. The presence of excessive abdominal fat in broilers has a direct impact on meat processing, resulting in lower slaughter rates and reduced consumer interest, thus affecting economic profitability [30,31]. The experiment found that replacing soybean meal in feed with ZBM significantly improved the dressed percentage, half-eviscerated weight rate, eviscerated weight rate, pectoral muscle rate, and leg muscle rate of broilers. Additionally, it also reduced the abdominal fat rate. The increase in muscle content and decrease in abdominal fat content in broilers often indicate enhanced protein digestibility and improved lipid metabolism [32]. All of these results indicated that substituting soybean meal in the diet with ZBM is not only an effective method but also positively contributes to the growth and development of broilers.

Serum biochemical indicators are dependable markers of health that reflect the nutritional, physiological, and pathological status of broiler chickens [33,34]. Alterations in serum parameters signify variations in the body’s metabolic and biochemical processes [35]. Serum proteins and their associated metabolic indicators, including total protein, albumin, globulin, and urea nitrogen, are commonly associated with protein absorption and metabolism [36]. In the experiment, the utilization of ZBM instead of soybean meal resulted in increased levels of albumin and globulin in serum. This increase suggests an improvement in protein digestion and utilization in broilers, which could be one of the reasons for the observed improvement in broiler production performance. Additionally, the ZBM group showed a significant reduction in serum levels of lipoprotein and total cholesterol, which is closely associated with the decrease in abdominal fat rate. Previous studies have indicated that a diet rich in polyunsaturated fatty acids can effectively decrease the triglyceride levels in broiler serum [32]. Furthermore, it has been observed that such a diet can positively influence fat metabolism and subsequently affect the levels of lipid metabolites in chicken serum [37]. ZBM demonstrates the ability to significantly improve lipid metabolism in broilers, thereby reducing total cholesterol content in serum and abdominal fat rate in broilers. Interestingly, the ZBM group showed a significant reduction in the activity of glutamic pyruvic transaminase and glutamic oxaloacetic transaminase in serum. Additionally, it increased the activity of superoxide dismutase and the content of immunoglobulin G in broilers, and led to a decrease in MDA content, which may be attributed to the inherent characteristics of ZBM, including its abundance in alkaloids, terpenoids, and flavonoids [12,13]. These compounds are believed to have a close association with the anti-inflammatory, antioxidant, and immune-enhancing properties of the feed [38]. The increase in antioxidant enzyme activity and the decrease in MDA content in serum indicate an improvement in the body’s antioxidant capacity, while the increase in immunoglobulin content suggests an enhancement in the immune function of broilers [39]. The decrease in serum ALT and AST activity suggests that substituting soybean meal with ZBM in the diet can improve liver health and positively impact production performance.

Compared with soybean meal, ZBM was detected to contain rich protein and amino acid content. Experiments in vitro have confirmed that ZBM exhibits significantly higher DM digestibility and extracorporeal digestible protein levels compared to soybean meal. Similarly, in the early growth stage, the digestion rates of DM, CP, and EN in the diet of ZBM groups broilers were found to be higher compared with those of the control group. The experiment hypothesizes that broilers in the early stages of growth and development have a relatively rapid metabolism, while the abundant alkaloids, terpenoids, and flavonoids in ZBM can help protect the intestines of broilers from foreign pathogens [12,13]. The environment of broiler intestines was significantly improved, leading to an enhancement in nutrient utilization efficiency; then, improvement was evident in the growth performance indicators [40,41]. Compared with the ZBM-1 group, the ZBM-2 group exhibited a decrease in the utilization of DM, CP, and EN in the diet at 42 days. However, there was no significant difference observed in these factors between the ZBM-2 group and the CON group, which suggests that substituting 10% soybean meal with ZBM did not have a detrimental effect on the nutrient utilization efficiency of broilers.

## 5. Conclusions

Overall, ZBM has confirmed the feasibility of replacing soybean meal in broiler chickens through both in vivo and vitro experiments. The use of 5–10% ZBM resulted in significant improvements in the growth and development of broilers, as well as the increased digestibility and utilization of nutrients. The experiment will serve as a valuable reference for the application of ZBM as a substitute for soybean meal in broilers, and it offers new methods for reducing and replacing soybean meal.

## Figures and Tables

**Figure 1 animals-14-01420-f001:**
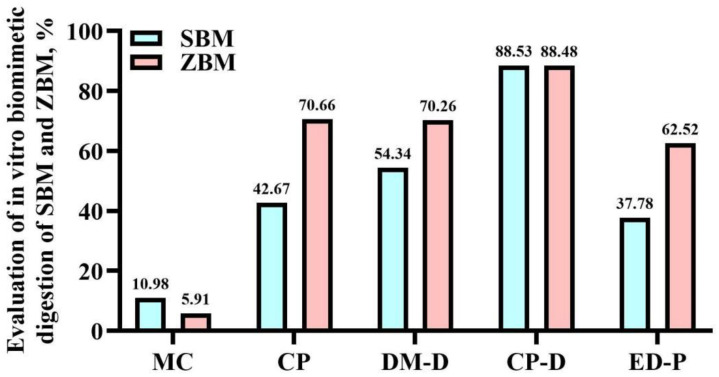
The vitro biomimetia digestion of SBM and ZBM. SBM—soybean meal; ZBM—*Zanthoxylum bungeanum* seed meal; MC—moisture content; CP—crude protein; MD-D—DM digestibility; CP-D—CP digestibility; ED-P—extracorporeal digestible protein.

**Figure 2 animals-14-01420-f002:**
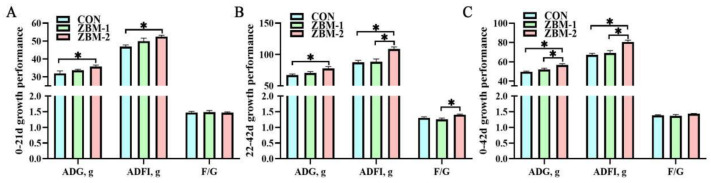
Effects of ZBM substitute for soybean meal in the diet on growth performance of broiler chickens. ADG—average daily gain; ADFI—average daily feed intake; F/G—feed-to-gain ratio. (**A**) 0–21 d growth performance; (**B**) 22–42 d growth performance; (**C**) 0–42 d growth performance. ZBM—*Zanthoxylum bungeanum* seed meal. In the ZBM-1 group, ZBM replaces 5% soybean meal in the diets; in the ZBM-2 group, ZBM replaces 10% soybean meal in the diets. Differences between treatments are displayed by * *p* < 0.05.

**Figure 3 animals-14-01420-f003:**
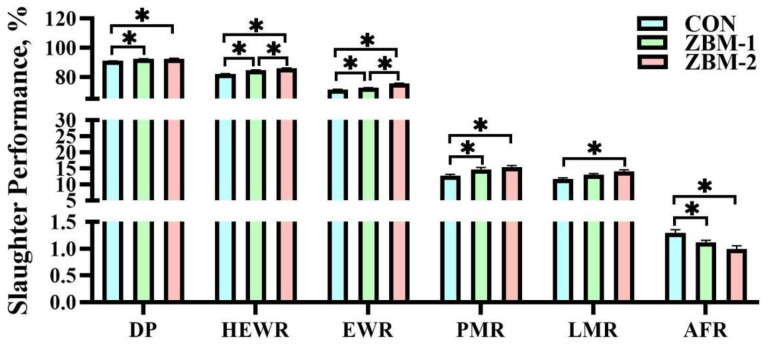
Effects of ZBM substitute for soybean meal in the diet on slaughter performance of broiler chickens. DP—dressed percentage; HEWR—half-eviscerated weight rate; EWR—eviscerated weight rate; PMR—pectoral muscle rate; LMR—leg muscle rate; AFR—abdominal fat rate; ZBM—*Zanthoxylum bungeanum* seed meal. In the ZBM-1 group, ZBM replaces 5% soybean meal in the diets; in the ZBM-2 group, ZBM replaces 10% soybean meal in the diets. Differences between treatments are displayed by * *p* < 0.05.

**Figure 4 animals-14-01420-f004:**
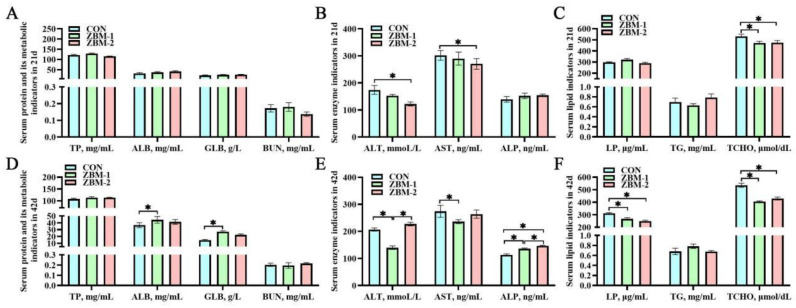
Effects of ZBM substitute for soybean meal in the diet on serum biochemical indicators of broiler chickens. TP—total protein; ALB—albumin; GLB—globulin; BUN—urea nitrogen; ALT—glutamic pyruvic transaminase; AST—glutamic oxaloacetic transaminase; ALP—alkaline phosphatase; LP—lipoprotein; TG—triglyceride; TCHO—total cholesterol. (**A**) Serum protein and its metabolic indicators in 21 d; (**B**) serum enzyme indicators in 21 d; (**C**) serum lipid indicators in 21 d; (**D**) serum protein and its metabolic indicators in 42 d; (**E**) serum enzyme indicators in 42 d; (**F**) serum lipid indicators in 42 d. ZBM—*Zanthoxylum bungeanum* seed meal. In the ZBM-1 group, ZBM replaces 5% soybean meal in the diets; in the ZBM-2 group, ZBM replaces 10% soybean meal in the diets. Differences between treatments are displayed by * *p* < 0.05.

**Figure 5 animals-14-01420-f005:**
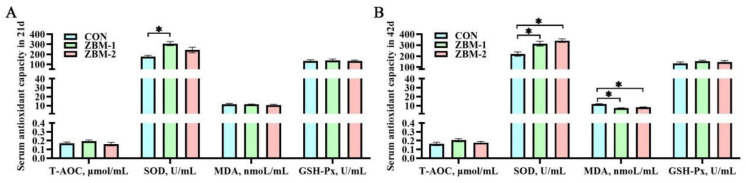
Effects of ZBM substitute for soybean meal in the diet on serum antioxidant capacity of broiler chickens. T-AOC—total antioxidant capacity; SOD—superoxide dismutase; MDA—malondialdehyde; GSH-Px—glutathione peroxidase. (**A**) Serum antioxidant capacity in 21 d; (**B**) serum antioxidant capacity in 42 d. ZBM—*Zanthoxylum bungeanum* seed meal. In the ZBM-1 group, ZBM replaces 5% soybean meal in the diets; in the ZBM-2 group, ZBM replaces 10% soybean meal in the diets. Differences between treatments are displayed by * *p* < 0.05.

**Figure 6 animals-14-01420-f006:**
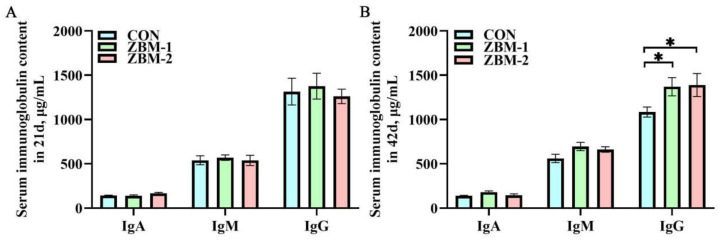
Effects of ZBM substitute for soybean meal in the diet on serum immunoglobulin content of broiler chickens. IgA—immunoglobulin A; IgM—immunoglobulin M; IgG—immunoglobulin G. (**A**) Serum immunoglobulin content in 21 d; (**B**) serum immunoglobulin content in 42 d. ZBM—*Zanthoxylum bungeanum* seed meal. In the ZBM-1 group, ZBM replaces 5% soybean meal in the diets; in the ZBM-2 group, ZBM replaces 10% soybean meal in the diets. Differences between treatments are displayed by * *p* < 0.05.

**Figure 7 animals-14-01420-f007:**
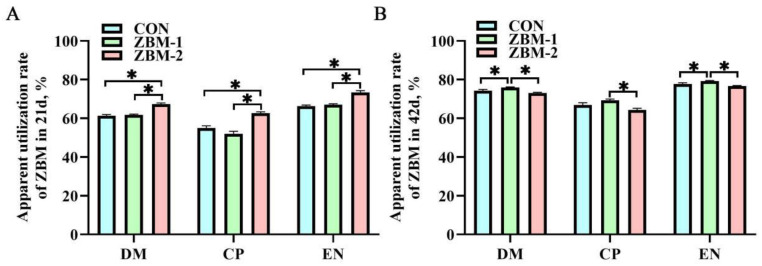
Effects of ZBM substitute for soybean meal in the diet on the apparent utilization rate of nutrients in broiler chickens. DM—dry matter; CP—crude protein; EN—energy. (**A**) Apparent utilization rate of ZBM in 21 d; (**B**) apparent utilization rate of ZBM in 42 d. ZBM—*Zanthoxylum bungeanum* seed meal. In the ZBM-1 group, ZBM replaces 5% soybean meal in the diets; in the ZBM-2 group, ZBM replaces 10% soybean meal in the diets. Differences between treatments are displayed by * *p* < 0.05.

**Table 1 animals-14-01420-t001:** The content of chemical composition and amino acid in SBM and ZBM.

Items	SBM	ZBM
Chemical composition, %		
Dry matter	88.30	91.94
Crude protein	46.32	70.66
Crude fat	1.87	3.0
Crude fibber	3.30	2.4
Crude ash	4.72	6.7
Indispensable amino acid, g/100 g		
Lysine	2.87	2.06
Methionine	0.61	1.26
Arginine	3.23	7.57
Histidine	1.18	1.55
Threonine	1.70	2.13
Valine	2.24	3.23
Leucine	3.49	4.85
Phenylalanine	2.32	2.61
Isoleucine	2.22	2.41
Tryptophan	0.49	0.88
Dispensable amino acid, g/100 g		
Alanine	1.96	2.43
Glycine	1.84	3.19
Glutamic acid	8.11	17.09
Proline	2.37	2.71
Serine	1.78	3.53
Cysteine	0.64	1.73
Aspartic acid	5.03	6.49
Tyrosine	1.60	2.82
Total	43.68	68.54

**Table 2 animals-14-01420-t002:** Ingredients and nutrient content of experimental diets (%, as-is basis).

Items	Contents					
	1–21 d			22–42 d		
	CON	ZBM-1	ZBM-2	CON	ZBM-1	ZBM-2
Corn	52.10	58.93	61.32	54.91	61.37	66.81
46% Soybean meal	33.43	23.56	14.45	27.20	17.63	8.20
ZBM	0.00	5.00	10.00	0.00	5.00	10.00
Soybean oil	3.06	0.79	0.00	4.09	1.87	0.00
Wheat bran	2.00	2.00	2.00	2.00	2.00	2.00
DDGS	5.00	5.00	5.00	8.00	8.00	8.00
NaCl	0.32	0.32	0.32	0.32	0.32	0.32
CaHPO_4_	1.73	1.78	1.87	1.44	1.50	1.56
Limestone	1.39	1.45	1.49	1.18	1.24	1.29
*L*-Lysine	0.34	0.51	0.66	0.32	0.48	0.64
*DL*-Methionine	0.30	0.29	0.28	0.26	0.25	0.24
*L*-Threonine	0.00	0.04	0.11	0.00	0.06	0.13
*L*-Tryptophan	0.00	0.00	0.01	0.00	0.00	0.01
Choline chloride	0.20	0.20	0.20	0.15	0.15	0.15
Zeolite	0.00	0.00	2.16	0.00	0.00	0.52
Premix	0.13	0.13	0.13	0.13	0.13	0.13
Nutrients ^1^						
Digestive energy, MJ/kg	12.55	12.55	12.55	12.97	12.97	12.97
Crude protein	20.46	20.50	20.47	18.45	18.43	18.42
Calcium, %	1.00	1.01	1.00	0.85	0.87	0.85
Phosphorus	0.70	0.66	0.62	0.64	0.61	0.61
Nonphytate phosphorus	0.45	0.45	0.45	0.42	0.42	0.42
Lysine, %	1.24	1.25	1.25	1.11	1.10	1.11
Methionine, %	0.62	0.62	0.61	0.56	0.55	0.55
Threonine, %	0.83	0.81	0.80	0.73	0.73	0.72
Methionine + Cysteine, %	0.95	0.95	0.95	0.84	0.85	0.85
Tryptophan, %	0.26	0.26	0.24	0.23	0.21	0.20

CON group, based diet; ZBM—*Zanthoxylum bungeanum* seed meal. In the ZBM-1 group, ZBM replaces 5% soybean meal in the diets; in the ZBM-2 group, ZBM replaces 10% soybean meal in the diets. ^1^ The numerical value represents the measured content of nutritional components.

## Data Availability

The raw data supporting the conclusions of this article will be made available by the authors, without undue reservation.

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
