# Peer review of "Dietary Replacement of Soybean Meal with Zanthoxylum bungeanum Seed Meal on Growth Performance, Blood Parameters, and Nutrient Utilization in Broiler Chickens"

_animals, 2024, doi:10.3390/ani14101420_

Round 1
Reviewer 1 Report
Comments and Suggestions for Authors
The article titled "Dietary Replacement of Soybean Meal with Zanthoxylum Bungeanum Seed Meal on Growth Performance, Blood Parameters, and Nutrient Utilization in Broiler Chickens" presents a meticulously conducted study that explores the feasibility of substituting soybean meal with Zanthoxylum bungeanum seed meal (ZBM) in the diets of broiler chickens. The manuscript commences with a concise introduction that provides a rationale for the need of investigating alternate protein sources, given the difficulties related to the availability and pricing of soybean meal. The methods part is comprehensive, offering a detailed explanation of the experimental design, covering the criteria for selection and the measurements of growth performance, blood parameters, and nutrients utilization. The findings are carefully presented, employing suitable statistical analysis and accompanied by supporting figures. The discussion provides a comprehensive analysis of the findings, examining the consequences of substituting soybean meal with ZBM on broiler performance and nutrient consumption. The paper is carefully crafted, following scientific writing rules and presenting a thoroughly substantiated position. In summary, this manuscript provides a significant contribution to the field by investigating the potential of ZBM as a substitute protein source in broiler chicken diets.
There are some comments on the enclosed file.

English Language is fine
Author Response
Response to Reviewer 1
Question 1: move this part to the subtitle 2.2. "Animals and diets"
Reply: Thank you for your suggestion. The modifications have been completed as required.
Question 2: Use the appropriate verb tense
Reply: Thank you for your suggestion. The modifications have been completed as required. (Line 27 and 30)
Question 3: Provide definitions for the rate "do you mean the relative percentage to the body wight of birds?
Reply: Yes. The method commonly used is the terminology and measurement statistical methods for poultry production performance (NY/T 823-2020). Compare slaughter performance indicators with pre slaughter weight. (Line 136)
Question 4: Provide definitions for the assessed carcass characteristics
Reply: Thanks for your advice. We usually refer to the terminology and measurement statistical methods for poultry production performance (NY/T 823-2020). (Line 140-141)
Question 5: L7177-178: Define abbreviations.
Reply: The definitions of DM-D and CP-D have been shown in the annotations of Figure 1. (Line 180-182)
Question 6: L209: Define abbreviations.
Reply: The definitions of HEWR and EWR have been shown in the annotations of Figure 1. (Line 218-220)
Question 7: L312: change to "1-2%"
Reply: Thank you for your suggestion. The modifications have been completed as required. (L312)
Question 8: line 331-358: divide this paragraph into separated two or three paragraphs.
Reply: Thank you for your feedback, which aligns with my original idea. But, i believe there is a reciprocal relationship among serum indicators, and isolating just 1-2 indicators may lead to brevity in the paragraphs. Therefore, i plan to consolidate the serum indicators and concentrate on analyzing the most significant ones.
Thank you for the valuable feedback received from the reviewers. We have carefully considered your requirements and made detailed modifications accordingly. In addition, we have also provided our own opinions on some issues. If you believe that any further modifications are necessary, please do not hesitate to contact us. We will we will quickly correct these issues.
Reviewer 2 Report
Comments and Suggestions for Authors
This is an interesting experiment to evaluate the energy and nutritional potential of a regional alternative feed ingredient to replace soybean meal in chicken feed. It would be interesting to determine the digestibility of amino acids in the test food, which would allow formulating diets based on these values ​​(composition of digestible amino acids). But in this study, the authors formulated the diets based on total amino acid composition, as recommended by the NRC (1994). Although this has limitations, the study is an important as na initial step in order to evaluate the potential of ZBM as a chicken feed ingredient.
Here are some questions and some notes requiring clarification or more detailed descriptions.
L_53: There is actually no shortage of soybean meal in the world! By the way, to consider in the INTRODUCTION, what is the production potential of this alternative feed ingredient (ZBM)? Is there really a possibility regarding considerable quantity/volume in order to replace part of the soybean meal in commercial chicken production? What is the production potential of ZBM?
L-89: But not metabolizable energy whose values ​​used were lower than the NRC.
L_85-92: What is the purpose of this experiment? This should be clear here. Based on the Results (Figures 2 to 6), these diets were designed to perform the "Growth Performance" test (described in 2.4). Therefore, it is suggested to insert this description (Animals and Diets) in the description of the Growth Performance test.
L_92-93: Need to detail how the value of Metabolizable Energy was estimated based on the chemical composition.
L_116-128: Is there any literature regarding the “Biomimetic Digestion Measurement Procedure for Total Digestible Carbohydrates in Chicken Feed” method? Could the authors detail the methodology used regarding incubation time and temperature in each of the test steps?
L_156: What was the inclusion (% or mg/kg) of TiO2?
Figure 1. How was the “Extracorporeal digestible protein” determined?
Fig. 7. There is a clear interaction between the DIET factors (or treatments) and AGE for the three variables evaluated. Using CP as an example, while at 21 days ZBM-2 was superior to ZBM-1, at 42 days the opposite occurred (ZBM-1 was superior to ZBM-2). This was also observed for DM and EM, especially when considering the two treatments containing ZBM (ZBM-1 and ZBM-2). Please consider this observation in the discussion, seeking to comment on a possible explanation.
Author Response
Response to Reviewer 2
Question 1: This is an interesting experiment to evaluate the energy and nutritional potential of a regional alternative feed ingredient to replace soybean meal in chicken feed. It would be interesting to determine the digestibility of amino acids in the test food, which would allow formulating diets based on these values (composition of digestible amino acids). But in this study, the authors formulated the diets based on total amino acid composition, as recommended by the NRC (1994). Although this has limitations, the study is an important as na initial step in order to evaluate the potential of ZBM as a chicken feed ingredient.
Reply: Thank you for your recognition of this experiment. In this experiment, the control group (CON group) of chicks were provided with a basic die formulated to meet or exceed the standards set by NRC (1994). This is a very distressing issue, and there is currently no clear standard to replace NRC 1994 and NY-T 2004. Each breed of broiler chickens also has certain differences in the utilization rate of feed nutrients, and we can only design experiments based on our own experience.
Question 2: L_53: There is actually no shortage of soybean meal in the world! By the way, to consider in the INTRODUCTION, what is the production potential of this alternative feed ingredient (ZBM)? Is there really a possibility regarding considerable quantity/volume in order to replace part of the soybean meal in commercial chicken production? What is the production potential of ZBM?
Reply: I completely agree with your viewpoint. However, in China, acquiring soybean meal can be challenging due to various factors. The volatility of soybean meal prices significantly impedes the progress of China's livestock and poultry industry. Therefore, it is essential to utilize feed options that are readily available and have high production in China as substitutes for soybean meal, in order to mitigate the risks associated with fluctuating soybean meal prices.
I am honored to discuss this issue with you and I am very excited about it. The corn-soybean meal diet is widely recognized as crucial in poultry farming. Despite China's vast territory and rich resources, some regions could benefit significantly from utilizing local raw materials as substitutes for soybean meal. China's substantial annual consumption of soybean meal is primarily imported due to limited domestic land resources and insufficient large-scale cultivation of beans, leading to over 80% reliance on foreign soybeans. This heavy dependence on imports poses operational and economic pressures on the feed and breeding industries. The rising cost of soybean meal and feed raw materials further exacerbates economic challenges. Decreasing soybean meal usage can help reduce feed costs and enhance breeding industry efficiency. In response, China's Ministry of Agriculture and Rural Affairs has initiated an effort to reduce and replace soybean meal in feed, aiming to decrease the proportion of soybean meal in feed from 14.5% in 2022 to less than 13% by 2025. Additionally, China is actively exploring alternative protein feed resources, such as grain processing by-products and microbial protein, to bolster protein feed supply. This experiment was conducted within this context.
Question 3: L-89: But not metabolizable energy whose values used were lower than the NRC.
Reply: Yes, thank you very much for your feedback.
Question 4: L_85-92: What is the purpose of this experiment? This should be clear here. Based on the Results (Figures 2 to 6), these diets were designed to perform the "Growth Performance" test (described in 2.4). Therefore, it is suggested to insert this description (Animals and Diets) in the description of the Growth Performance test.
Reply: This study aims to explore the impact of substituting some soybean meal with ZBM on the growth performance of broilers. The chemical and amino acid composition of ZBM was analyzed, and in vitro digestion was conducted to confirm its digestibility. Subsequently, broiler experiments were carried out to assess its practical application in broiler diets.
Question 5: L_92-93: Need to detail how the value of Metabolizable Energy was estimated based on the chemical composition.
Reply: The chemical composition and amino acid composition of ZBM were analyzed to estimate its metabolic energy based on the energy contribution of these components, which is in line with our understanding of raw materials in basic feed.
Question 6: L_116-128: Is there any literature regarding the “Biomimetic Digestion Measurement Procedure for Total Digestible Carbohydrates in Chicken Feed” method? Could the authors detail the methodology used regarding incubation time and temperature in each of the test steps?
Reply: Thanks for your feedback. Regarding "Biomimetic Digestion Measurement Procedure for Total Digestible Carbohydrates in Chicken Feed", we marked the company name (Hunan Zhongben Intelligent Technology Development Co.; Ltd.; Hunan, China), which can be clearly explained on its official website.
Question 7: L_156: What was the inclusion (% or mg/kg) of TiO2?
Reply: Use 0.4% titanium dioxide for this experiment. The modifications have been made in the text.
Question 8: Figure 1. How was the “Extracorporeal digestible protein” determined?
Reply: Thank you for your advise. In vitro simulated digestion processes involve using simulated digestive fluids, such as gastric and intestinal fluids, to replicate the digestive environment of the body. These fluids contain digestive components like acids and enzymes that mimic the digestive processes occurring in the stomach and intestines. Proteins are broken down into smaller peptide segments and amino acids as they are exposed to these simulated digestive fluids. By assessing the extent of protein degradation during simulated digestion, researchers can evaluate the in vitro digestibility of the protein.
Question 9: Fig. 7. There is a clear interaction between the DIET factors (or treatments) and AGE for the three variables evaluated. Using CP as an example, while at 21 days ZBM-2 was superior to ZBM-1, at 42 days the opposite occurred (ZBM-1 was superior to ZBM-2). This was also observed for DM and EM, especially when considering the two treatments containing ZBM (ZBM-1 and ZBM-2). Please consider this observation in the discussion, seeking to comment on a possible explanation.
Reply: In response to this issue, we are currently engaged in extensive indicator mining, analyzing pertinent amino acid utilization pathways and conducting cecal microbiota sequencing. However, overall, the nutrient utilization efficiency of ZBM in broiler chickens is still higher than that of soybean meal.
Thank you for the valuable feedback received from the reviewers. We have carefully considered your requirements and made detailed modifications accordingly. In addition, we have also provided our own opinions on some issues. If you believe that any further modifications are necessary, please do not hesitate to contact us. We will we will quickly correct these issues